# Nanoparticle-Mediated Delivery of Deferasirox: A Promising Strategy Against Invasive Aspergillosis

**DOI:** 10.3390/bioengineering11111115

**Published:** 2024-11-05

**Authors:** Sydney Peppe, Moloud Farrokhi, Evan A. Waite, Mustafa Muhi, Efthymia Iliana Matthaiou

**Affiliations:** 1Department of Immunology and Microbial Disease, Albany Medical College, Albany, NY 12208, USA; peppes25@mail.wlu.edu (S.P.); farrokm@amc.edu (M.F.); muhim@amc.edu (M.M.); 2Albany Medical College, Washington and Lee University, Lexington, VA 24450, USA; 3Albany Medical College, Rochester Institute of Technology, Rochester, NY 14623, USA

**Keywords:** invasive aspergillosis (IA), antifungal resistance, nanoparticles (NPs), deferasirox (DFX), poly(lactic-co-glycolic acid) (PLGA), iron chelators, safe and effective treatment

## Abstract

Background: Invasive aspergillosis (IA) is a deadly fungal lung infection. Antifungal resistance and treatment side effects are major concerns. Iron chelators are vital for IA management, but systemic use can cause side effects. We developed nanoparticles (NPs) to selectively deliver the iron chelator deferasirox (DFX) for IA treatment. Methods: DFX was encapsulated in poly(lactic-co-glycolic acid) (PLGA) NPs using a single emulsion solvent evaporation method. The NPs were characterized by light scattering and electron microscopy. DFX loading efficiency and release were assessed spectrophotometrically. Toxicity was evaluated using SRB, luciferase, and XTT assays. Therapeutic efficacy was tested in an IA mouse model, assessing fungal burden by qPCR and biodistribution via imaging. Results: DFX-NPs had a size of ~50 nm and a charge of ~−30 mV, with a loading efficiency of ~80%. Release kinetics showed DFX release via diffusion and bioerosion. The EC50 of DFX-NPs was significantly lower (*p* < 0.001) than the free drug, and they were significantly less toxic (*p* < 0.0001) in mammalian cell cultures. In vivo, NP treatment significantly reduced *Af* burden (*p* < 0.05). Conclusion: The designed DFX-NPs effectively target and kill *Af* with minimal toxicity to mammalian cells. The significant in vivo therapeutic efficacy suggests these NPs could be a safe and effective treatment for IA.

## 1. Introduction

*Aspergillus* spp. can cause a range of respiratory diseases called “aspergillosis” that extend from hypersensitivity to invasive disease [1]. The immune system plays a crucial role in the risk of aspergillosis. Individuals with weakened immune systems are more susceptible to the disease [2]. However, the factors that determine the severity and type of aspergillosis are poorly understood. Recent epidemiological studies estimate that aspergillosis affects about 15 million people annually [3]. Over 2,113,000 people develop invasive aspergillosis (IA), with a crude annual mortality of 85.2% [3]. Despite the growing concern, fungal infections receive very little attention and resources, leading to unmet research and development needs [4]. Recognizing the increasing fungal infection burden, the World Health Organization (WHO) recently published the first-ever list of priority fungal pathogens and ranked infectious fungi into critical, high, and medium priority groups based on the severity of disease and treatment challenges [5]. *Aspergillus fumigatus* (*Af*), a ubiquitous opportunistic airborne mold, was ranked as one of the four critical priority fungi [4]. Aspergillosis treatment and management are challenged by limited access to quality diagnostics and therapeutics as well as the emergence of antifungal resistance. Antifungal resistance has major implications for human health. It generally leads to prolonged therapy and hospital stays, and an increased need for expensive and often highly toxic second-line antifungal medicines [6]. Agricultural use is the main cause of the rising rates of azole-resistant *Aspergillus fumigatus* infections. Azole-resistant strains of *Aspergillus* have been reported around the globe, but the true prevalence and rate of triazole resistance remain underestimated and partly unknown [7,8]. Furthermore, environmental changes contribute to *Aspergillus’* global burden. There is an increase in (i) the bioavailability of *Aspergillus* conidia in the air due to increased growth of fungi as temperatures rise and (ii) the conidia dispersal due to storms. Additionally, several pollutants are also known to cause direct damage to the lung epithelia and the local microbiome that provides an innate barrier to infection; this damage creates a favorable micro-environment for *Aspergillus* infection [9,10]. Finally, pulmonary viral infections seem to be associated with aspergillosis. Viral invasive aspergillosis (VIA) has emerged in intensive critical care (ICU) settings following the recent influenza H1N1 and Coronavirus Disease 2019 (COVID-19) pandemics. VIA seems to differ from IA in terms of pathogenicity and clinical presentation [11,12]. Considering the current challenges in IA treatment and the incidence increase in this fungal infection, there is an urgent need to develop novel therapeutic approaches to effectively treat IA.

Iron chelators are being explored as a potential therapeutic strategy for treating *Af* infections, including invasive aspergillosis. Iron is essential for the growth and virulence of *Af* [13]. Iron chelators work by binding to iron, making it unavailable to the fungus, thereby inhibiting its growth and biofilm formation [14]. Studies have shown that common iron chelators like deferiprone (DFP) and deferasirox (DFX), can inhibit biofilm formation in a dose-dependent manner [14].

In clinical settings, the use of iron chelators requires careful consideration of the specific chelator (deferoxamine is bioavailable to *Aspergillus* spp. since the fungus can use it as siderophore) and the susceptibility of the *Aspergillus* strain. Individual assessments are often necessary to determine the most effective treatment [14]. The ongoing research on the therapeutic potential of iron chelators is focused on understanding the complex interactions between iron metabolism and fungal virulence, as well as developing new iron chelators that are more effective and have fewer side effects [15]. In addition, some studies have explored the use of iron chelators in combination with traditional antifungal drugs (azoles and polyene antibiotics) [16,17,18,19]. For example, combining DFP with antifungal agents has shown synergistic effects, improving treatment outcomes [20]. Preliminary results suggest that iron chelation therapy could be a viable adjunct to conventional antifungal treatments. Clinical trials aim to determine the optimal dosing, safety, and efficacy of iron chelators in patients with invasive aspergillosis, while there is a Phase 2 clinical trial using radiolabeled deferoxamine for the visualization of pulmonary *Aspergillus* infection [21].

DFX is a Food and Drug Administration (FDA)-approved iron chelator for the treatment of chronic iron overload due to blood transfusions in patients aged 2 years and older, and for chronic iron overload in non-transfusion-dependent thalassemia syndromes in patients aged 10 years and older [1]. The systemic administration of DFX can lead to several adverse events (AEs), some of which can be serious. Common AEs include skin reactions and gastrointestinal issues such as nausea, vomiting, diarrhea, and stomach pain. Severe AEs include renal failure, liver failure, often fatal gastrointestinal hemorrhage, severe allergic reactions, and vision and hearing problems. IA usually affects immunocompromised individuals [22]. While there is no FDA guidance on the use of DFX in immunocompromised patients, it is likely that the AEs of DFX will be more pronounced in these individuals due to their condition. Thus, the targeted delivery of DFX to the site of *Af* infection would be a better therapeutic strategy. DFX is a good candidate to treat IA since it has a high affinity for ferric iron (Fe^3^⁺), which is essential for the growth and metabolism of *Af*, the primary pathogen in IA [23,24]. Furthermore, its safety profile is well documented; thus, it is a viable option for repurposing in the treatment of IA. By binding to ferric iron, DFX deprives the fungus of this critical nutrient, inhibiting its growth and proliferation.

It is well known that drug nanodelivery offers several advantages over traditional systemic administration. There is a plethora of FDA-approved nanoparticles (NPs) (doxil, abraxane, onivyde, ambisome, vyxeos, and marqibo) that are now preferred treatments over the conventional drugs. NPs can improve the solubility and stability of drugs, enhancing their absorption and effectiveness [25]. Additionally, NPs can be designed to release their payload in a controlled manner over time. This sustained release can maintain therapeutic drug levels for extended periods, reducing the frequency of dosing and improving patient compliance [26]. By concentrating the drug at the target site and reducing systemic exposure, nanodelivery systems can lower the risk of the adverse effects and toxicity associated with high systemic drug levels [25,27]. Encapsulation within nanoparticles can protect drugs from degradation by enzymes or other biological processes, extending their shelf life and effectiveness [28]. Furthermore, the behavior of NPs in the respiratory system is influenced by their size, shape, and surface properties. NPs with a size of 50 nm can penetrate deep into the lungs and may accumulate there [29]; thus, targeted delivery in the lungs can be achieved by NP size manipulation.

The objective of this study is to develop NPs for the targeted delivery of DFX in the lungs to treat IA. DFX loaded into poly(lactic-co-glycolic acid) (PLGA) NPs represents a novel therapeutic approach for IA with unique advantages compared to the existing methods for drug delivery and treatment, particularly in the context of antifungal therapies. (i) PLGA nanoparticles can be engineered to target the lungs [30], enhancing the accumulation of DFX at the infection site. This targeted delivery helps in maximizing the therapeutic effect while minimizing systemic side effects. (ii) PLGA NPs could provide a controlled and sustained release of DFX, ensuring a steady therapeutic concentration over an extended period [31]. This reduces the need for frequent dosing and helps maintain effective drug levels. (iii) Encapsulating DFX in PLGA nanoparticles protects the drug from degradation and enhances its stability, improving its shelf life and efficacy [32]. (iv) The use of nanoparticles can enhance the solubility of poorly soluble drugs like DFX, improving its overall efficacy [33]. This is a significant advantage over traditional formulations that may struggle with solubility issues. (v) Additionally, the unique properties of DFX-PLGA NPs may help overcome drug resistance. By utilizing a novel mechanism of action, these nanoparticles can potentially re-sensitize resistant fungal strains to the existing antifungals, offering a new strategy for managing difficult infections. (vi) PLGA safely degrades into lactic acid and glycolic acid, which are naturally metabolized by the body [34] (vii) PLGA NPs can improve the pharmacokinetic profile of DFX, enhancing its absorption, distribution, and retention in the body [35]. These attributes make DFX PLGA NPs a promising strategy for treating IA, potentially improving patient outcomes and reducing the burden of this serious fungal infection.

## 2. Materials and Methods

### 2.1. Nanoparticle Formulation

For the formulation of PLGA NPs, we employed a single emulsion solvent evaporation technique. Briefly, DFX (10 mg/mL) (HY-17359 MedChemExpress, Monmouth Junction, NJ, USA) was dissolved in Dimethyl Sulfoxide (DMSO) (BP231100, Fisher Bioreagents, PA USA). A total of 100 mg of PLGA acid-terminated lactide/glycolide 50:50, Mw 24,000–38,000 (719870, Sigma-Aldrich, MA, USA) was dissolved in oil phase 10 mL ethyl acetate ACS 99.5+% (AA31344AP, Thermo Scientific Chemicals, MA, USA). Then, DFX was added to the oil phase. A total of 200 mg of Poloxamer 188 (K4894, Sigma-Aldrich, MA, USA) was dissolved in 20 mL of double distilled deionized water (976610LHP, Aqua Solutions, MA, USA). The oil phase was added dropwise to the water phase while stirring. The solution was sonicated on ice at 60% power for 1 m with 10 s break (4 cycles) using a probe sonicator (VCX 130; Sonics and Materials, Inc., Newtown, CT, USA). Then, the oil phase was eliminated by evaporation under reduced pressure (150 mbar) using a rotary evaporator (30 rpm) (Hei-VAP Advantage ML/G5B; Heidolph North America, Grove Village, IL, USA); the solvent evaporation rate was approximately 1 mL/min. Drug-free PLGA NPs were formulated using the same procedure. The NP solution was stored for up to one week at 4 °C before being used in experiments.

### 2.2. Particle Size and Zeta Potential Analyses

To determine the size and zeta potential of the NPs, 20 μL of each sample was diluted to 1 mL of distilled water and the size and zeta potential of the NPs were analyzed using a Zetasizer Nano-ZS (Malvern Instruments, Malvern, UK).

### 2.3. Transmission Electron Microscopy (TEM) and Scanning Electron Microscopy (SEM)

For morphological characterization, the formulated NPs were evaluated by TEM and SEM. For TEM, 10 μL of the NP sample was suspended in 1 mL of distilled water. One drop of this suspension was placed over a carbon 400 mesh TEM grid and allowed to dry. Images were visualized at 120 KeV at indicated magnifications on a Tecnai 12 microscope (FEI, Hillsboro, OR, USA) equipped with a Gatan, Inc. (Pleasanton, CA, USA) 896 2.2.1 US1000 camera. For SEM, 10 μL of the NP sample was suspended in 1 mL of distilled water. One drop of this suspension was placed on a premium plain glass microscope slide (125444, Fisherbrand lab equipment, Waltham, MA, USA). The samples were allowed to dry on the slide and were imaged using a Hitachi SU-70 microscope (Hitachi, Tokyo, Japan) with 1 nm resolution at 15 kV.

### 2.4. Drug Entrapment (DE%) and Drug Loading (DL%) Efficiencies

The NP content of DFX was estimated by the ultraviolet (UV) spectrophotometric method. The DFX-loaded NPs were dissolved in dichloromethane (DCM) > 99.9% (34856 Sigma-Aldrich, MA, USA) (1 mg/mL) and kept on a shaker at 37 °C for 72 h until the complete release of the entrapped drug. To ensure complete dissolution, the NP solution was sonicated for 30 s in an ice bath. After sonication, the solution was clear and free of aggregates. Then, the samples were centrifuged at 10,000× *g* for 5 min at 4 °C to extract the drug present in the solution. The supernatant (700 μL) was collected and 125 µL was added to 175 µL imaging solution (2,6-Dichloroquinone-4-chloroimide 98%-AC155100250 Thermo Scientific, MA, USA) and Borate Buffer pH 9.5–104,032, Rica Chemical Company, Arlington, TX, USA). The absorbance was measured by a plate reader (BioTek Cytation 5, Agilent, CA, USA) at a wavelength of 658 nm [36]. Drug concentration was calculated based on the DFX standard curve (0, 2, 4, 6, 8, and 10 mg/mL Appendix A. We used the following equations to calculate drug entrapment and loading efficiencies:DL% = ((Dt mass − Df mass)/NP mass) ×100
DE% = ((Dt mass − Df mass)/Dt mass) ×100
Df: free drug; Dt: total drug.

### 2.5. Release Kinetics Analysis

The in vitro release kinetics of DFX from PLGA NPs was determined using FBS fetal bovine serum (FBS) (A5670801, Gibco, NY, USA) at various pHs (pH 4.4, 5.4, 6.4, and 7.4) at 37 °C. The NPs (25 mg) were dispersed in 5 mL of the FBS buffer and divided into equal aliquots (1 mL each). These tubes were kept on a shaker at 37 °C and 150 rpm. At designated time intervals (1, 2, 4, 8, 12, 24, 48, and 72 h), these tubes were taken from the shaker and centrifuged at 10,000× *g* at 4 °C for 5 min. Then, the supernatant was removed and the amount of released drug was estimated as above using an imaging reagent and absorbance at 658 nm. The release data were analyzed using the Korsmeyer–Peppas model (M_t_/M_∞_ = kt^n^, M_t_ = drug release at time t, M_∞_ = total amount of drug, k = release rate constant, and n = release exponent).

### 2.6. Mammalian Cell Culture

Cell lines H292 (human lung epithelial cells, CRL-1848) and Raw264.7 (mouse macrophage cells, TIB-71) were purchased from the American Type Culture Collection (ATCC, Manassas, VA USA). The cells were cultured onto well plates using the RPMI 1640 and DMEM media supplemented with 10% fetal bovine serum, 100 units/mL penicillin G, and 100 μg/mL streptomycin. The cells were maintained in a humidified incubator set at 37 °C with 5% CO_2_, ensuring optimal conditions for cultivation and experimentation.

### 2.7. Fungal Culture

*Aspergillus fumigatus Af*293 strain was cultured in potato dextrose agar (PDA) (BD 213400, BD Difco^TM^, Franklin Lakes, NJ, USA) Petri dishes for 7 days prior to harvesting conidia for experimental use. The *Af*293, also known as strain CBS 101355, was purchased by ATCC (MYA-4609).

### 2.8. Cytotoxicity Evaluation

#### 2.8.1. Mammalian Cells

The cytotoxicity of free DFX, empty NPs, and DFX NPs was evaluated in mammalian cell lines by the well-established sulforhodamine B (SRB) [37] assay using a commercially available kit (786-213, CytoScan^TM^ SRB Cell Cytotoxicity Assay, G Biosciences, St. Louis, MO, USA). Briefly, the cells were seeded in 96-well plates (50.000 cells/well). After 24 h, the cells were treated with designated concentrations (0 mM-control, 2 mM, 4 mM, 8 mM, and 16 mM) of DFX or equivalent of empty and DFX NPs for 24 and 48 h. At the end of incubation, the treated cells were fixed using 10% trichloroacetic acid for 60 min at 4 °C. After washing four times with distilled water, the plates were left to air-dry and then stained with SRB staining solution (0.057% *w/v* in 1% *v/v* acetic acid solution) for 30 min at room temperature. The cells were washed four times with 1% acetic acid solution. After drying, the cells were solubilized with 200 μL of 10 mM Tris base solution (pH 10.5) on a shaker for 30 min. The absorbance was measured using the BioTek Cytation 5 plate reader at 570 nm. The cell cytotoxicity was calculated using the following equation: % Cytotoxicity = (100 × (Cell Control − Experimental)) ÷ (Cell Control) [38]. The software GraphPad Prism 10 version 10.4.0 (621)was used to determine the EC50.

#### 2.8.2. Fungal Cultures

The inhibition of fungal proliferation was evaluated by the 2,3-Bis(2-methoxy-4-nitro-5-sulfophenyl)-2H-tetrazolium-5-carboxanilide (XTT) assay [39,40,41]. We used a commercially available XTT assay kit (CyQUANT^TM^ XTT Cell Viability Assay, Invitrogen, USA). Briefly, the conidia were cultured at a seeding density of 10^6^ conidia/well onto 96-well plates, and treatments were added as described above. After 24 h and 48 h, 50 µL of XXT labeling mixture was added and the cells were incubated for an additional 4 h at 5% CO2 and 37 °C. The absorbance was measured using a BioTek Cytation 5 plate reader at 450 nm and 660 nm. Specific absorbance was calculated using the following formula: Specific Absorbance = [Abs_450nm_(Sample) − Abs_450nm_(Blank)] − Abs_660nm_(Sample) [42]. Growth inhibition was calculated via the following equation: %growth inhibition = [1 − (Abs_Sample_/Abs_Control_)] × 100. The software GraphPad Prism 10 was used to determine the EC50.

### 2.9. In Vivo Biodistribution Model

For this study, we initially used C57Bl/6 mice (Appendix A, but finally, we selected a nude (nu/nu) mouse model. The nu/nu mice are widely used as a standard model for biodistribution studies involving fluorophore detection due to their unique characteristics [43,44]. (i) The lack of fur in nu/nu mice reduces light scattering and absorption, which enhances the clarity and accuracy of fluorescence imaging. This is particularly important for detecting and quantifying the distribution of fluorophores within the body [45]. (ii) The nu/nu mice immunodeficiency ensures that there is a minimal immune response to the injected agents, leading to more consistent and reproducible results in biodistribution studies [46]. PLGA NPs loaded with indocyanine green dye (IGC) (I2633, Cardiogreen powder, Sigma-Aldrich, USA) were injected (100 μL/injection) intravenously (IV) in female athymic nu/nu mice (strain NU/J #002019, 6 weeks old, The Jackson Laboratory, Bar Harbor, ME, USA) (n = 5/group). Infrared detection images were taken before injection and immediately after injection, 20 min, 1 h, 2 h, 4 h, 6 h, 12 h, 24 h, 48 h, and 72 h after injection. If 72 h after injection no signal was detected, the mice were sacrificed, organs were harvested (the brain, heart, lungs, thyroid, kidneys, spleen, liver, fallopian tubes, and ovaries), and infrared detection images of the organs were taken using the Li-COR Pearl Impulse instrumentation. ImageJ was used to quantify the signal of each organ in the images taken.

### 2.10. Neutropenic Mouse Model of IA

Both male and female 6-weeks-old C57BL/6 mice were used (strain C57BL/6J #000664, The Jackson Laboratory). G*Power was used to calculate the required sample size for an alpha of 0.05 and a power of 0.8; a sample size n = 5/group was sufficient for detecting a meaningful effect. Neutropenia in mice was established via cyclophosphamide solution intraperitoneal (IP) injections (150 mg/kg) two days before injections and three days after infection. On day 0, the mice were infected via the inhalation of *Af*293 conidia. Briefly, the mice were placed in an inhalation chamber and 10^8^/mL conidia were nebulized for 30 min to ensure adequate exposure. Immediately after infection on day 0 and daily until day 7, the mice were treated with 1× sterile phosphate-buffered saline (PBS) (AAJ61196AP, Thermo Scientific Chemicals, MA, USA) (control), DFX 20 mg/kg (free drug), empty NPs (NP control), and DFX NPs 20 mg/kg intravenously. On day 8, the mice were sacrificed, and the lungs were collected for further analyses. The mice were monitored daily for changes in weight and signs of distress. Supportive care including hydration and nutritional support was provided as needed.

### 2.11. qPCR to Determine Fungal Burden

The infected mouse lungs were cut into small pieces using surgical scissors. PBS (1 mL per 100 mg tissue) was added, and the tissue was homogenized using a dounce homogenizer followed by a rotor–stator homogenizer (D1000 Handheld Homogenizer, Benchmark Scientific, Edison, NJ, USA). Lung homogenates were centrifuged to remove debris, and the supernatant was collected. DNA was extracted using the DNeasy Blood & Tissue Kit (69506, Qiagen Inc., Germantown, MD, USA) according to the manufacturer’s instructions. *Aspergillus* detection was performed using the GeneProof *Aspergillus* PCR Kit (60ASP002025, Alpco Ltd., Salem, NH, USA) targeting the multicopy sequence of the intergenic spacer ITS2/28S rDNA. *Af*293 DNA served as the positive control, and a reaction without DNA was the negative control. The PCR cycle included an initial denaturation at 95 °C for 2 min, followed by 40 cycles of 95 °C for 5 s and 60 °C for 40 s. Amplification curves were analyzed using a real-time PCR machine. The presence of *Af*293 DNA was indicated by the fluorescence signal from the probes. Fungal burden was quantified by comparing the sample’s Ct values to a standard curve of known Af293 DNA concentrations and normalized to host DNA using mouse GAPDH as a housekeeping gene.

### 2.12. Study Approval

All the animal studies were approved by the IACUC and Albany Medical College.

### 2.13. Statistical Analysis

All the statistical analyses were performed using the GraphPad Prism 10 software. *p*-value was set at *p* > 0.5 for statistically significant differences.

## 3. Results

### Nanoparticle Characterization

All the DFX PLGA NP formulations had a size range varying from 20 to 70 nm as measured by DLS with an average size diameter of 50 nm. The surface charge (zeta potential) of the NPs was measured by ELS. The surface charge of the nano formulations slightly varied (~±1 mV) with an average of −32.9 mV (Figure 1).

To comprehensively characterize the DFX-loaded PLGA NPs, we used Transmission Electron Microscopy (TEM) and Scanning Electron Microscopy (SEM). The TEM provided high-resolution micrographs that allowed the detailed visualization of the internal structure and crystallinity of the NPs. The SEM micrographs enabled the examination of the surface morphology, topography, and distribution of the NPs, complementing the data collected by TEM. Based on the TEM and SEM micrographs, the DFX NPs exhibited a homeodispersed size distribution (~50 nm), a spherical shape, and a smooth, unblemished surface (Figure 1).

To estimate the amount of DFX encapsulated within the PLGA NPs relative to the total amount of DFX used in the formulation process, we determined the loading efficiency (DL%) of the DFX NPs. To measure the percentage of the initial DFX successfully encapsulated within the nanoparticles, we calculated the entrapment efficiency (DE%) of the DFX NPs. Evaluating both efficiencies is important to ensure that a significant portion of DFX has been incorporated into the PLGA NPs and that the formulation process is effective. According to our calculations described in the methods section, the loading and entrapment efficiencies (DL% and DE%) of the DFX NPs were 77.9% and 79.2%, respectively (Table 1).

We tested the DFX release of NPs in serum at pH levels 4.4, 5.4, 6.4, and 7.4 to mimic physiological conditions. Specifically, pH 7.4 represents the normal pH of blood and extracellular fluids [47]. pH 6.4 is characteristic of inflamed and infected tissues [48]. pH 5.5 reflects the pH of endosomes and lysosomes within cells, where nanoparticles might be internalized [49]. Finally, pH 4.4 simulates highly acidic conditions found in some cellular compartments and in the stomach [50,51]. Our study showed that 20% of the drug was released within the first 2 h, and approximately 80% was released after 48 h. The release of the drug from NPs appeared to be somewhat pH-dependent, with the highest release at pH 7.4 (Figure 2A). The drug release profile at pH levels 4.4–7.4 fitted the Korsmeyer–Peppas model, with drug release at pH 4.4 best fitting the model (R^2^ = 0.9771). The release exponent (n) ranged from 0.6959 (pH 7.4) to 0.7455 (pH 4.4), suggesting drug release due to a combination of diffusion and bioerosion (Figure 2B).

We tested the effect of DFX NP treatment on cell viability in the lung epithelial cell line H292 and the macrophage cell line Raw264.7, as the bronchi primarily consist of epithelial cells and macrophages [52]. In the Raw264.7 cells, free DFX significantly reduced cell viability (*p* < 0.0001) compared to DFX NPs, suggesting that DFX nanodelivery has a prophylactic effect in this cell population (Figure 3A). In the H292 cells, free DFX did not reduce cell viability; however, DFX NPs were significantly less toxic (*p* = 0.02) compared to free DFX, indicating a prophylactic effect (Figure 3B). Both the DFX and DFX NP treatments were toxic to the *Af*293 cultures compared to the control and control NPs. However, DFX NPs were significantly more toxic (*p* < 0.0001) to the *Af*293 cultures compared to free DFX, suggesting effective drug delivery to the fungus (Figure 3C). Finally, empty PLGA NPs had no effect on the cell or *Af*293 cultures. The effective concentration 50 (EC50) was calculated for DFX and DFX NPs in the Raw264.7, H292, and *Af*293 cells (Figure 3B,D,F). The EC50 values for DFX and DFX NPs were compared, revealing that the EC50 for DFX NPs was significantly lower in *Af*293 compared to both mammalian cell lines (Figure 3G).

We evaluated the biodistribution of these NPs in vivo by formulating indocyanine green (ICG)-loaded PLGA NPs and administering these NPs intravenously (IV) to female nude mice. We monitored biodistribution using infrared imaging over 72 h. At 24 h, no signal was detected in the mice. The mice were sacrificed at 72 h, and organs were harvested and imaged for infrared signal (Figure 4A). We identified a very strong and significantly higher (*p* < 0.0001) ICG signal in the lungs compared to the other organs. Some ICG signal was also detected in the liver (Figure 4B). These results imply that NPs are delivered and sustained in the lungs, suggesting the targeted delivery of the NPs in the lungs, where IA manifests.

Finally, we tested the efficacy of the DFX NPs in a mouse model of IA (Figure 5A). We used PBS and empty PLGA NPs as controls and compared the therapeutic efficacy of free DFX to DFX NPs after 7 days of treatment post *Af*293 infection. We found that the DFX NPs significantly (*p* = 0.0001) reduced the *Af*293 burden in the lungs compared to free DFX. Furthermore, there was no significant burden between the uninfected mice and infected mice treated with DFX NPs, indicating that the fungus was almost cleared from the lungs after the DFX NP treatment (Figure 5B). Additionally, we monitored the animals’ weight throughout the treatments as an indirect method to evaluate potential toxicity. The weight monitoring data demonstrated that DFX NPs had no impact on the weight of the mice, whereas the mice treated with PBS, control NPs, and DFX showed weight loss (Figure 5C). These results imply that the DFX NPs were not toxic.

## 4. Discussion

We formulated PLGA NPs loaded with DFX with a size of ~50 nm and a surface charge of ~−30 mV. *Aspergillus* cell wall consists of negatively charged components such as glucans, chitin, and glycoproteins [53,54]. These components contribute to the overall negative charge of the cell wall, which can influence interactions. The mammalian cell membrane is also negatively charged. If the NPs were positively charged, they would likely have electrostatic interactions with both mammalian cells and fungal hyphae, thus negative charge was preferred. The DFX loading efficiency was high ~80%. High loading efficiency means that more DFX can be delivered to the target site, potentially increasing its therapeutic effect. Furthermore, with more drugs loaded into each nanoparticle, the frequency of dosing can be reduced, increasing the efficiency of use. Additionally, by delivering a higher concentration of the drug directly to the target site, systemic side effects can be minimized. PLGA degrades through the hydrolysis of its ester bonds, which is influenced by the pH of the surrounding environment. In acidic conditions, the hydrolysis rate increases, leading to faster degradation of the polymer and, consequently, quicker drug release. In neutral or basic conditions, the degradation is slower, resulting in a more sustained release. However, DFX has poor water solubility and is highly sensitive to pH changes. In acidic conditions, such as in gastric fluid, DFX is practically insoluble. This explains why most of the DFX was released at pH 7.4. Furthermore, this could explain why the NPs have a prophylactic effect in mammalian cells. After phagocytosis, the NPs are transferred to phagolysosomes, where the acidic pH reduces the drug’s solubility and effectiveness. The release exponent (n) at the Korsmeyer–Peppas model of drug release can provide information about how the DFX is being released from the NPs. n = 0.5 indicates that the drug is being released due to passive diffusion. 0.5 < n < 1 indicates that the drug is being released due to a combination of diffusion and bioerosion, while n = 1 indicates controlled release due to bioerosion. Since bioerosion happens in acidic environments and DFX is insoluble in acidic conditions, we aimed for a drug release that would combine drug diffusion and bioerosion. Furthermore, 20% of the DFX was released in serum within the first 2 h. The time it takes for NPs to reach the lungs after intravenous (IV) administration can vary based on several factors, including the size, surface properties, and composition of the nanoparticles. Generally, nanoparticles can reach the lungs within minutes after IV administration. Thus, we anticipate this initial passive DFX diffusion to have an impact on the lungs as well as a systemic effect. The bronchi, where aspergillosis is established, comprise epithelial cells and macrophages. Thus, we decided to evaluate the toxicity of the DFX NPs in epithelial cells and macrophages. NPs with a size of 50 nm are expected to be phagocytosed more easily compared to NPs of bigger size. However, NPs with a negative surface charge, such as −30 mV, tend to be less readily phagocytosed. This could have been attributed to DFX NP's prophylactic effect observed in mammalian cells compared to free DFX. While it was not within the scope of this work to investigate the internalization of the NPs by *Aspergillus* hyphae, we observed that DFX nanodelivery had higher toxicity to the *Af*293 culture compared to free DFX, implying that the DFX NPs can interact with the fungal hyphae and deliver DFX. However, additional research is needed to define these interactions and confirm the NP's internalization. Furthermore, DFX NPs at a concentration of 8 mM were completely toxic to the fungus while they had no toxicity to mammalian cells, indicating that (i) the nanodelivery of DFX is more effective compared to free DFX; (ii) via nanodelivery, we can use lower concentrations of DFX and still achieve high fungal toxicity. Our biodistribution study demonstrated that PLGA NPs with a size of ~50 nm can accumulate and be retained in the lungs after 72 h. This targeted delivery is essential since it can enhance the efficacy of the drug at the site of infection while reducing toxicity to other tissues. On the other hand, some ICG signal was also observed in the liver, and the NPs retained in the liver might induce liver dysfunction and toxicity. While our data indicate targeted delivery, additional studies are needed to confirm drug clearance and toxicity.

This high lung retention was unexpected. The reticuloendothelial system (RES) typically directs nanoparticles (NPs) to the liver and spleen [55,56,57], but instability and aggregation can alter this distribution, leading to lung accumulation [58]. We monitored NP precipitation via DLS size measurement during the in vitro release assays. We found that the size of DFX PLGA NPs significantly (**** *p* < 0.00001) increased over time in FBS at pH 7.4 (from ~51 nm to ~730 nm at t = 72 h) (Appendix A). This size increase indicates that NPs precipitate in serum. This nanoparticle precipitation could have contributed to the observed lung accumulation. When nanoparticles aggregate and precipitate, they form larger particles that are more likely to be deposited and become trapped in the narrow capillaries of the lungs [59,60].

Nanoparticle (NP) aggregates that can stay in lung capillaries typically range in size from 100 nm to 1 µm [61,62]. The lung capillaries are very narrow, with diameters around 5–10 µm, which means that larger aggregates can become trapped and obstruct blood flow [63,64,65]. Thus, NP aggregates that are larger than 1–2 μm (µm) are likely to cause lung embolism. Smaller aggregates, especially those under 1 µm, are less likely to cause embolism as they can pass through the capillaries more easily. However, as the size of the aggregates increases, the risk of them becoming trapped and causing embolism also increases [66,67].

While the NP aggregates in FBS were ~720 nm after 72 h and within the “safe” size range, the impact of NP precipitation should be examined further. Furthermore, there are several other factors that could explain the higher accumulation of 50 nm PLGA NPs in the lungs compared to the liver. (i) The lung’s extensive capillary network acts as a filter, capturing particles of this size [68,69,70]. (ii) The IV-injected NPs encounter the lung capillaries before reaching the other organs and this increases the likelihood of their retention in the lungs [71,72,73,74]. (iii) The lungs have a high concentration of phagocytic cells, such as alveolar macrophages, which can uptake and retain nanoparticles [75,76,77,78,79]. (iv) The surface charge and hydrophobicity of PLGA nanoparticles can influence their interaction with lung tissue; hydrophobic particles are more likely to be retained in the lungs [80,81,82,83]. Thus, 50 nm sized NPs are more likely to be trapped in the lung capillaries due to their size.

Finally, we observed significant therapeutic efficacy of the DFX NPs in the mouse model of IA. The DFX NPs seem to clear the fungal infection from the lungs as a monotherapy. The observation of significant therapeutic efficacy of DFX NPs (nanoparticles) in a mouse model of invasive aspergillosis (IA) is important for several reasons. First, it suggests a promising new treatment option for IA, which is often difficult to manage with conventional antifungal therapies. The ability of DFX NPs to clear fungal infections from the lungs as a monotherapy could lead to more effective patient outcomes. Additionally, nanoparticle-based therapies typically allow for targeted delivery, which can minimize side effects compared to systemic treatments. This targeted approach may enhance patient compliance and improve the quality of life for those affected by IA. Moreover, fungal infections, particularly in immunocompromised patients, can develop resistance to the existing antifungal drugs. The efficacy of DFX NPs may provide an alternative mechanism of action, potentially overcoming these resistance challenges. This finding also lays the groundwork for further research to explore the mechanisms of action, optimal dosing, and potential applications in human patients, paving the way for future clinical trials. If successful in humans, this approach could be adapted for other fungal infections, contributing to a broader arsenal of treatments against fungal diseases, which are increasingly recognized as significant health threats. Overall, these findings could significantly impact the management of invasive fungal infections, especially in vulnerable populations.

Furthermore, we anticipate that using these NPs in combination with known antifungals will increase their therapeutic efficacy. DFX NPs could serve as a promising combination treatment with current antifungals for several reasons. First, they may enhance the efficacy of the existing therapies by delivering the drug directly to the site of infection, increasing its concentration where it is needed most. This targeted approach could also help overcome resistance, as many fungal strains have developed mechanisms to evade conventional antifungals. Additionally, the use of nanoparticles can reduce the toxicity associated with systemic treatments, minimizing exposure to healthy tissues and improving patient compliance. There is also the potential for synergistic effects, where the combination of DFX NPs and the existing antifungals proves more effective than either treatment alone. Importantly, if DFX NPs can re-sensitize azole-resistant strains to antifungals, this would represent a significant advancement in managing difficult infections. Overall, integrating DFX-loaded NPs with the current antifungal therapies could greatly improve treatment outcomes, particularly for patients facing resistant strains. For the scope of this work, we have not tested these NPs in azole-resistant *Aspergillus* strains. It is important to investigate whether these NPs have toxic effects on azole-resistant strains and if the DFX-NPs can re-sensitize these strains to azoles.

In conclusion, our study demonstrates that DFX-loaded nanoparticles (NPs) offer a promising approach for treating invasive aspergillosis (IA) through a combination of drug diffusion and bioerosion. The initial rapid release of DFX in serum, coupled with the targeted delivery and retention of NPs in the lungs, suggests a dual impact on both local and systemic levels. The observed prophylactic effect of DFX NPs in mammalian cells, along with their higher toxicity to *Aspergillus* hyphae compared to free DFX, highlights the potential of nanodelivery to enhance therapeutic efficacy while minimizing toxicity. Our findings indicate that DFX NPs can effectively clear fungal infections in a mouse model of IA, and further research is warranted to explore their impact on azole-resistant strains and potential synergistic effects with the existing antifungals. Additionally, while some liver accumulation was noted, further studies are needed to confirm drug clearance and assess long-term toxicity. Overall, DFX NPs represent a significant advancement in targeted antifungal therapy, offering a more effective and safer alternative to conventional treatments.

## Figures and Tables

**Figure 1 bioengineering-11-01115-f001:**
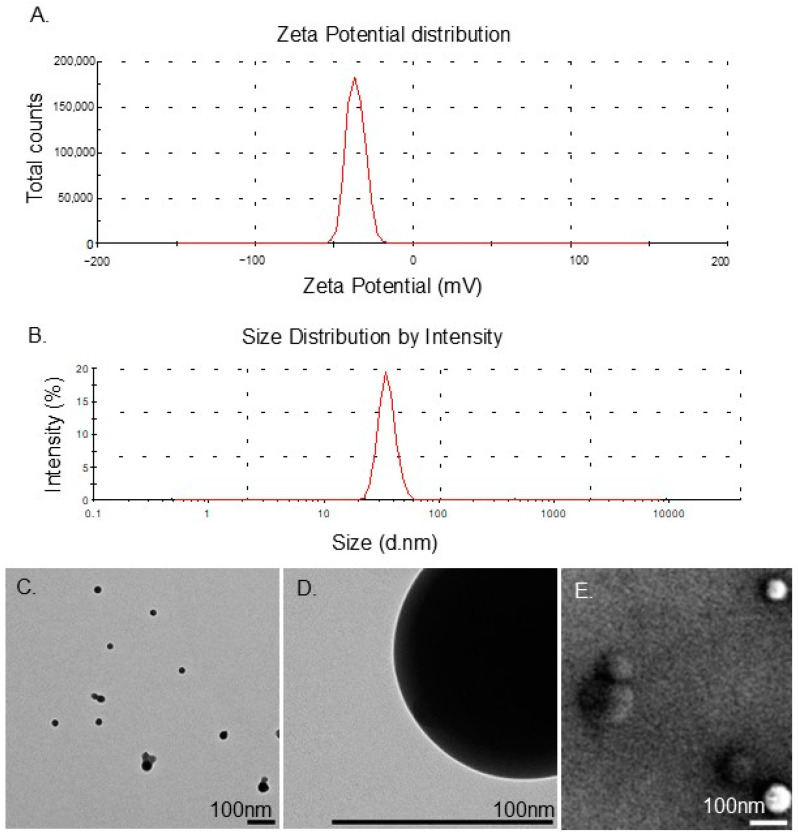
Charge and size distribution of DFX NPs and the TEM and SEM micrographs of the DFX-loaded PLGA NPs. (**A**). Zeta potential of DFX-loaded PLGA NPs. (**B**). Size distribution of DFX-loaded PLGA NPs. (**C**). TEM micrograph showing shape and size distribution. (**D**). TEM micrograph showing the absence of cracks in the NP’s surface. (**E**). SEM micrograph showing smooth spherical surface and size distribution.

**Figure 2 bioengineering-11-01115-f002:**
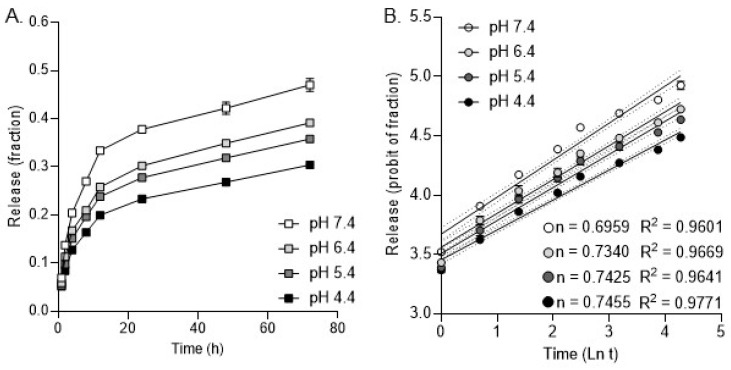
The DFX release data profile from PLGA NPs in FBS. (**A**). DFX release data over 72 h in pH 4.4, 5.4, 6.4, and 7.4. (**B**). Release data fit with the Korsmeyer–Peppas drug release kinetic model.

**Figure 3 bioengineering-11-01115-f003:**
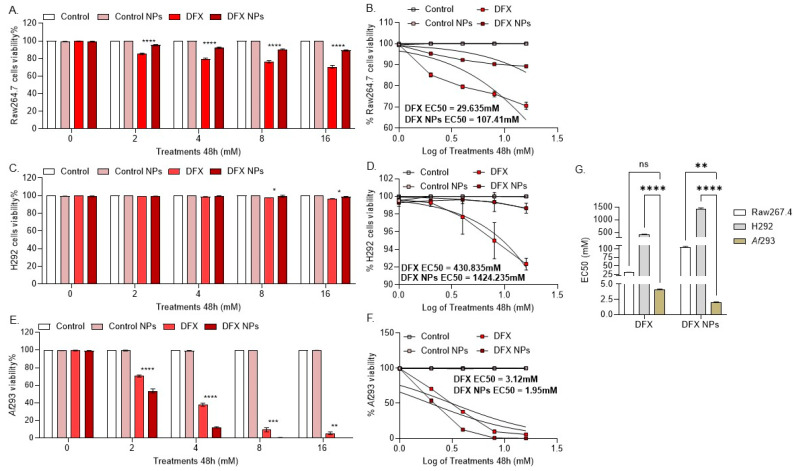
Viability assays in mammalian cells and *Af* cultures at 48 h. (**A**). SRB assay data in Raw264.7 cells (**** *p* < 0.0001). (**B**). Calculated EC50 of treatments in Raw264.7 cells. (**C**). SRB assay data in H292 cells (* *p* < 0.05). (**D**). Calculated EC50 of treatments in H292 cells. (**E**). XTT assay data in *Af*239 culture (** *p* = 0.0016, *** *p* = 0.0003, and **** *p* < 0.0001). (**F**). Calculated EC50 of treatments in *Af*293 culture. (**G**). Comparison of EC50 in Raw264.7, H292, and *Af*293 cultures (ns = not significant, ** *p* = 0.0016; **** *p* < 0.00001).

**Figure 4 bioengineering-11-01115-f004:**
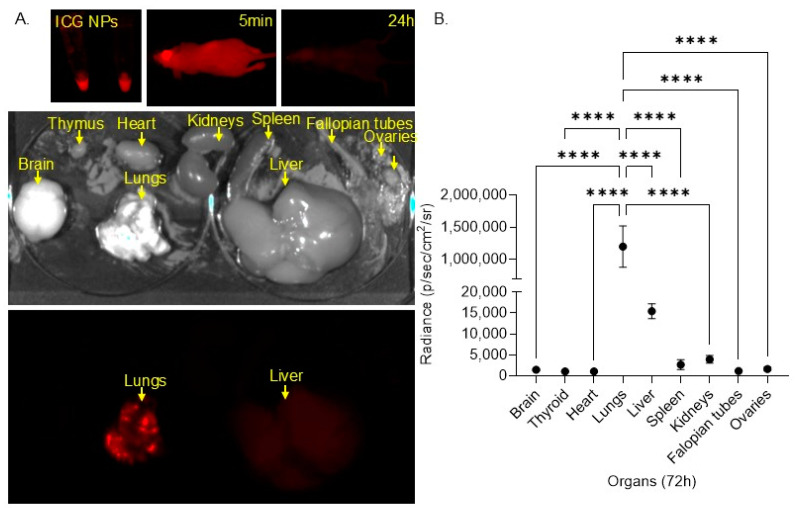
Biodistribution study in nude mice. (**A**). Infrared images of the NP formulation, the mice after 5 min and 24 h, and the harvested organs (the brain, thymus, heart, lung, kidneys, spleen, liver, cervix, and ovaries) 72 h after treatment with PLGA NPs loaded with indocyanine green (ICG) 0.5 nmol. (**B**). Quantification of infrared signal from each organ after 72 h (**** *p* < 0.0001).

**Figure 5 bioengineering-11-01115-f005:**
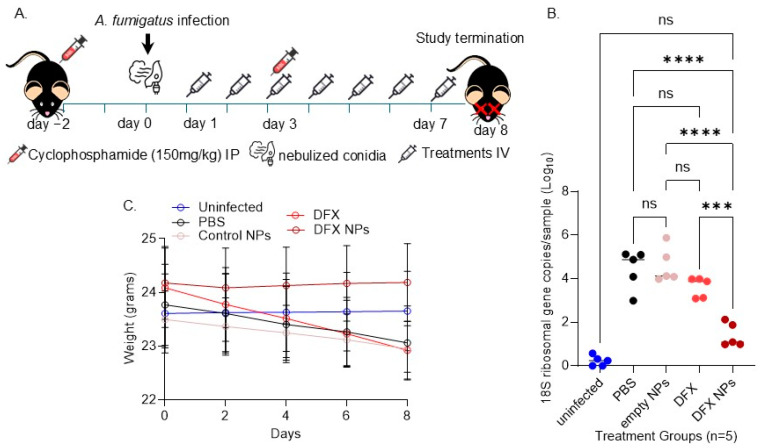
In vivo efficacy of the treatments in the neutropenic mouse model of IA. (**A**). Schematic mouse model and treatments. (**B**). Fungal CFUs estimated via 18S RNA qPCR (ns = not significant, *** *p* = 0.0001; **** *p* < 0.0001). (**C**). Weight monitoring.

**Table 1 bioengineering-11-01115-t001:** DL% and DE% of the DFX-loaded NPs.

Equation	DFX NPs
DL% = ((Dt mass − Df mass)/NP mass) × 100	77.9%
DE% = ((Dt mass − Df mass)/Dt mass) × 100	79.2%

**Abbreviations:** DE%, drug entrapment efficiency; Df, free drug; DL%, drug loading efficiency; Dt, total drug; NP, nanoparticle; DFX, deferasirox.

## Data Availability

The data presented in this study are available on request from the corresponding author. The data are not publicly available due to privacy restrictions.

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
