# Peer review of "Nanoparticle-Mediated Delivery of Deferasirox: A Promising Strategy Against Invasive Aspergillosis"

_bioengineering, 2024, doi:10.3390/bioengineering11111115_

Round 1

Reviewer 1 Report

Comments and Suggestions for Authors

Comments:

  1. Ensure all abbreviations are defined upon first use (e.g., SRB, XTT, AF, ...).
  2. Include relevant references in lines 32-33.
  3. Revise lines 36-39 for grammatical accuracy.
  4. Add appropriate references in lines 75-78.
  5. Revise and clarify lines 90-92.
  6. Provide detailed characteristics of materials, polymers, and drugs in Section 2, including molecular weight, purity, etc.
  7. Specify how prepared NPs were harvested and stored for further experiments in the "Nanoparticle formulation" section.
  8. Describe the Scanning Electron Microscopy (SEM) method for morphological characterization.
  9. Define "SHK" in line 149.
  10. Include relevant references for "at a wavelength of 658nm".
  11. Add references for equations in "Cytotoxicity evaluation" and "Fungal cultures".
  12. Clarify discrepancies between results mentioned in lines 261-262 and Figure 3A.
  13. Include amount of EC50 as stated in the Fungal cultures section in the results.
  14. Correct caption labels B and C of Figure 7.
  15. Include details on measuring fungal burden in the lungs using qPCR targeting the 18S ribosomal gene in the "qPCR to determine fungal burden" section.
  16. Discuss reasonable explanations for targeted drug delivery and accumulation of developed nanoparticles in lung tissue in the Discussion section.
  17. Expand the discussion with detailed interpretation of results and integration of additional references.

Comments on the Quality of English Language

The quality of english language is good.

Author Response

Thank you for the opportunity to revise our manuscript. We appreciate the insightful comments and suggestions.

  1. Ensure all abbreviations are defined upon first use (e.g., SRB, XTT, AF, ...).
  • We searched the document, and all abbreviations are defined upon first use.

2. Include relevant references in lines 32-33.

  • We included references in lines 32-33.

3. Revise lines 36-39 for grammatical accuracy.

  • We adjusted the sentence to assure clarity and grammar.

4. Add appropriate references in lines 75-78.

  • Appropriate references were added in lines 75-78.

5. Revise and clarify lines 90-92.

  • Lines 90-92 were revised for clarity.

6. Provide detailed characteristics of materials, polymers, and drugs in Section 2, including molecular weight, purity, etc.

  • We provided details of all materials used for the experiments described.

7. Specify how prepared NPs were harvested and stored for further experiments in the "Nanoparticle formulation" section.

  • We provided information on how NPs were collected and stored before experimental use.

8. Describe the Scanning Electron Microscopy (SEM) method for morphological characterization.

  • SEM method was described.

9. Define "SHK" in line 149.

  • SHK was corrected.

10. Include relevant references for "at a wavelength of 658nm".

  • Reference was added.

11. Add references for equations in "Cytotoxicity evaluation" and "Fungal cultures".

  • References were added.

12. Clarify discrepancies between results mentioned in lines 261-262 and Figure 3A.

  • We rewrote the drug release kinetics section. We also discussed the DFX release in acidic environment at the discussion section.

13. Include amount of EC50 as stated in the Fungal cultures section in the results.

  • EC50 calculations and values were included.

14. Correct caption labels B and C of Figure 7.

  • We corrected the caption labels.

15. Include details on measuring fungal burden in the lungs using qPCR targeting the 18S ribosomal gene in the "qPCR to determine fungal burden" section.

  • We added the details of the kit used to identify aspergillus CFUs using qPCR.

16. Discuss reasonable explanations for targeted drug delivery and accumulation of developed nanoparticles in lung tissue in the Discussion section.

  • We added an explanation on why NPs showed targeted delivery to the lungs at the discussion section.

17. Expand the discussion with detailed interpretation of results and integration of additional references.

  • We expanded the discussion and added additional references.

We are grateful for the constructive feedback and believe that the revisions have strengthened our manuscript. We look forward to your further comments.

Reviewer 2 Report

Comments and Suggestions for Authors

In this study, the authors demonstrated DFX-loaded nanoparticles offer a promising approach for treating invasive aspergillosis (IA). The research direction is meaningful, especially for innovative treatments for IA. The use of nanoparticles to deliver iron chelators to meet the challenges of antifungal treatment has high innovation and clinical translation potential. The article is well-structured and clearly describes the study background, methods, results and discussion.  The experimental steps for the preparation, characterization and evaluation of nanoparticles  in the article are clear and detailed. However, the article still has many shortcomings that need to be improved:

1. It is recommended that when introducing the advantages of nanoparticles, more specific examples of their application in similar antifungal treatments should be listed, and the uniqueness of DFX-PLGA nanoparticles compared with existing methods should be clearly stated.

2. For the preparation conditions of nanoparticles, describe the key parameters in each step (such as ultrasound intensity, solvent evaporation rate, etc.) as detailed as possible to facilitate other researchers to reproduce the experiment.

3. The figures in the article need to be further improved. The authors should refer to figures in published high-level articles.

4. Although the article mentioned the use of mouse models, the number of mice in each group and the statistical considerations of the experimental design should be more clearly stated.

5. For biodistribution studies, the authors should use wild-type mice instead of immunodeficient nude mice, because the immune system can significantly affect the transport and distribution of nanoparticles in the body. In addition, the subsequent treatment experiments in this article used wild-type mice. 

6. The authors should explain why the nanoparticles are significantly enriched in the lungs, even more than in the liver.

7.It is recommended to more clearly point out the future research direction and the possibility of clinical transformation in the conclusion.

Author Response

Thank you for the opportunity to revise our manuscript. We appreciate the insightful comments and suggestions.

In this study, the authors demonstrated DFX-loaded nanoparticles offer a promising approach for treating invasive aspergillosis (IA). The research direction is meaningful, especially for innovative treatments for IA. The use of nanoparticles to deliver iron chelators to meet the challenges of antifungal treatment has high innovation and clinical translation potential. The article is well-structured and clearly describes the study background, methods, results and discussion.  The experimental steps for the preparation, characterization and evaluation of nanoparticles in the article are clear and detailed. However, the article still has many shortcomings that need to be improved:

  1. It is recommended that when introducing the advantages of nanoparticles, more specific examples of their application in similar antifungal treatments should be listed, and the uniqueness of DFX-PLGA nanoparticles compared with existing methods should be clearly stated.
    • To our knowledge there are no FDA-approved nanoparticles specifically for the treatment of invasive aspergillosis. However, research is ongoing in nanodelivery of antifungal agents. These nanoparticles aim to enhance the efficacy of existing treatments while minimizing side effects. In December 2023, the FDA approved isavuconazonium sulfate (Cresemba) for treating invasive aspergillosis and invasive mucormycosis, particularly in pediatric patients. While this approval does not pertain to nanoparticles, it highlights the ongoing efforts to improve treatment options for invasive fungal infections. The development of nanoparticle-based therapies could represent a significant advancement in managing invasive aspergillosis, especially as researchers explore their potential to overcome drug resistance and improve drug delivery. However, there are no similar antifungal treatments against IA to list. The uniqueness of this treatment is being discussed in the last section of the introduction.

  1. For the preparation conditions of nanoparticles, describe the key parameters in each step (such as ultrasound intensity, solvent evaporation rate, etc.) as detailed as possible to facilitate other researchers to reproduce the experiment.
    • We added additional information including the solvent evaporation rate at the nanoparticle formulation section. We did not use ultrasound.

  1. The figures in the article need to be further improved. The authors should refer to figures in published high-level articles.
    • The figures match the journal’s instructions; we reassessed the quality of the figures. Please advise if we need to correct specific elements in the figures.

  1. Although the article mentioned the use of mouse models, the number of mice in each group and the statistical considerations of the experimental design should be more clearly stated.
    • Statistical method for power calculation was included in the IA model method section.

  1. For biodistribution studies, the authors should use wild-type mice instead of immunodeficient nude mice, because the immune system can significantly affect the transport and distribution of nanoparticles in the body. In addition, the subsequent treatment experiments in this article used wild-type mice.
    • WT mice fur was not allowing us to see the NPs signal in the mouse. Thus, we decided to use mice without fur. However, we added at the supplementary figures the analysis of organs harvested from WT mice at 72h and there were no statistically significant differences between the WT and the nude mice.

  1. The authors should explain why the nanoparticles are significantly enriched in the lungs, even more than in the liver.
    • We discussed why NPs showed targeted delivery to the lungs in the discussion section.

  1. It is recommended to more clearly point out the future research direction and the possibility of clinical transformation in the conclusion.
    • Thank you for this recommendation. We now discuss future directions and clinical applicability of this therapy at the discussion section.

We are grateful for the constructive feedback and believe that the revisions have strengthened our manuscript. We look forward to your further comments.

Round 2

Reviewer 1 Report

Comments and Suggestions for Authors

Accept

Author Response

Thank you once again for your thoughtful review.

Reviewer 2 Report

Comments and Suggestions for Authors

Thanks to the authors for their thoughtful responses. They have addressed most concerns well, and the manuscript quality has improved significantly. However, one issue remains that still requires clarification:

The authors have not adequately explained why PLGA nanoparticles accumulated more in the lungs than in the liver. Typically, following injection, nanoparticles accumulate primarily in the liver and spleen due to the reticuloendothelial system (RES) activity in these organs. Based on my experience, I suspect that the instability of PLGA nanoparticles may lead to precipitation in the body, resulting in their entrapment within the narrow capillaries of the lungs. This instability might stem from the lack of hydrophilic blocks, such as PEG. Testing this hypothesis could involve monitoring particle size changes when the nanoparticles are mixed with serum.

Author Response

Comment: Thanks to the authors for their thoughtful responses. They have addressed most concerns well, and the manuscript quality has improved significantly. However, one issue remains that still requires clarification:

The authors have not adequately explained why PLGA nanoparticles accumulated more in the lungs than in the liver. Typically, following injection, nanoparticles accumulate primarily in the liver and spleen due to the reticuloendothelial system (RES) activity in these organs. Based on my experience, I suspect that the instability of PLGA nanoparticles may lead to precipitation in the body, resulting in their entrapment within the narrow capillaries of the lungs. This instability might stem from the lack of hydrophilic blocks, such as PEG. Testing this hypothesis could involve monitoring particle size changes when the nanoparticles are mixed with serum.

Response: Thank you very much for your thoughtful and constructive feedback. We appreciate your recognition of the improvements made to our manuscript. We are grateful for your continued engagement and for highlighting the remaining issue regarding the accumulation of PLGA nanoparticles in the lungs.

Several factors may have led to the PLGA nanoparticle aggregation and precipitation in vivo, such as the lack of stabilizing agents like PEG and interaction with serum proteins. However, smaller hydrophilic nanoparticles like the ones we formulated tend to be more stable in suspension. To test if the PLGA nanoparticles precipitated in serum at 37°C, we used the nanoparticle DLS size measurements from the in vitro release assay at pH 7.4. Please refer to the newly added supplementary figure. The nanoparticles in serum formed visible sediments, and the size of these aggregates was significantly larger than the original nanoparticles. These findings suggest possible nanoparticle precipitation.

This nanoparticle precipitation could have contributed to the observed lung accumulation. When nanoparticles aggregate and precipitate, they form larger particles that are more likely to be deposited and become trapped in the narrow capillaries of the lungs. As you have correctly pointed out, the reticuloendothelial system (RES) typically directs nanoparticles to the liver and spleen, but instability and aggregation can alter this distribution, leading to increased lung accumulation.

We have added a supplementary figure and text to the manuscript discussion section. Please review and let us know if the updated version provides an adequate explanation for the observed lung accumulation. Thank you.